# Evaluation Model of Remote Sensing Satellites Cooperative Observation Capability

**Zhonggang Zheng** [1,2,3,4,5] **Qingmei Li** [6,*] and **Kun Fu** [1,2,4]

1  Aerospace Information Research Institute, Chinese Academy of Sciences, Beijing 100094, China; Zhengzhonggang15@mails.ucas.edu.cn (Z.Z.); fukun@mail.ie.ac.cn (K.F.)
2  School of Electronic, Electrical and Communication Engineering, University of Chinese Academy of Sciences, Beijing 100049, China
3  Chinese Academy of Sciences, Beijing 100094, China
4  Key Laboratory of Network Information System Technology, Institute of Electronics, Chinese Academy of Sciences, Beijing 100094, China
5  Beijing Institute of Remote Sensing Information, Beijing 100089, China
6  College of Engineering, Peking University, Beijing 100871, China
*  Correspondence: qmli@stu.pku.edu.cn; Tel.: +86-150-0048-8037

**Abstract:** This paper proposed a new remote sensing observation capability evaluation model (RSOCE) based on analytic hierarchy process to quantitatively evaluate the capability of multi-satellite cooperative remote sensing observation. The analytic hierarchical process model is a combination of qualitative and quantitative analysis of systematic decision analysis method. According to the objective of the remote sensing cooperative observation mission, we decompose the complex problem into several levels and a number of factors, compare and calculate various factors in pairs, and obtain the combination weights of different schemes. The model can be used to evaluate the observation capability of resource satellites. Taking the optical remote sensing satellites, such as China's resource satellite series and GF-4, as examples, this paper verifies and evaluates the model for three typical tasks: point target observation, regional target observation, and moving target continuous observation. The results show that the model can provide quantitative reference and model support for comprehensive evaluation of the collaborative observation capability of remote sensing satellites.

**Keywords:** remote sensing; collaborative application; observation capability; evaluation

## 1. Introduction

Remote sensing satellites can achieve full-time, large-scale, and highly reliable global observations, which is an important means to obtain information of global hotspots and large-scale regional surveys. Remote sensing satellites with decentralized management usually provide emergency high-frequency observation services in the form of satellite maneuvering orbit or camera tilting. However, its economic cost is extremely high, the orbit cannot fully meet the requirements of global emergency support [1]. Due to the complexity of the remote sensing satellite system itself and the diversity of application requirements, there is a great deal of uncertainty in the establishment of evaluation standards [2,3], the direction of the evaluation results is often ambiguous, and the decision-making support role is not obvious. Therefore, it is necessary and urgent to systematically study the method of remote sensing satellite system effectiveness analysis and evaluation, and comprehensively use multiple remote sensing satellites to provide data support in emergency situations.

In order to provide emergency coverage to any location on Earth, it is important to ensure that the correct combination of satellites is used. This requires a system model that can fully discover and use existing sensor technology to meet target observation requirements and achieve optimal effect of resource utilization for different application services.

The performance evaluation of remote sensing satellites includes task analysis [4], index system construction, evaluation method selection, and so on. There are few literatures on the mission effectiveness evaluation of remote sensing satellites, which mainly focus on mission analysis, index system construction, or comprehensive evaluation method research [5]. Moreover, the comprehensive evaluation research of the remote sensing satellite system is not sufficient [6], but the evaluation of the whole space equipment system or satellite equipment system is mostly carried out [7]. In recent years, the research on the comprehensive evaluation of remote sensing satellite systems effectiveness is not sufficient, and most of them are for the whole aerospace equipment system or satellite equipment system [8–10].

There are two types of commonly used remote sensing satellite effectiveness evaluation systems. One is to build an evaluation index system starting from the demand description model, such as evaluating satellite imaging, from the three aspects of image quality, detection time, and detection area [11]. The other is to build a remote sensing application efficiency index system based on four types of indicators: spatial resolution, time resolution, spectral range, and spectral resolution [12]. The disadvantage of the method is that each indicator reflects an aspect of the system's capabilities, and is almost irrelevant to each other, and lacks comprehensive indicators to evaluate the system's capabilities as a whole.

The combination of analytic hierarchy process (AHP) and the Availability Dependability Capability (ADC) effectiveness model is more widely used to evaluate the effectiveness of remote sensing satellite information acquisition [13–15]. AHP is essentially an expert evaluation method, which deals with various decision-making factors in qualitative and quantitative ways, and has the advantages of being systematic, flexible and concise. It has been widely valued and applied in the field of complex system evaluation. AHP is a method for decision making that is widely used, particularly for group decision making in a high-stakes area. AHP helps decision-makers choose between options by providing a rational framework for structuring thinking around a decision. The basic principle of AHP is to decompose the problem into different factors and to organize these in a hierarchical clustering combination [16], to form a hierarchical and ordered "hierarchical structure model". Each factor is then given a quantitative weight, based on the judgement of multiple decision-makers and using mathematical methods to determine relative importance at each level [17]. Finally, by comprehensively calculating the weight of relative importance of factors at various levels, we can obtain the importance weight of the lowest level factors to the highest level, or rank the order of advantages and disadvantages, as the basis for the evaluation and selection of the scheme. This method can be used to quantitatively evaluate the system performance and mission effectiveness, and to compare and analyze a variety of system schemes, which is very suitable to for the evaluation of satellite cooperative capability.

We combine actual requirements and expert experience to propose a remote sensing observation capability evaluation model (RSOCE). We have conducted research on the target identification, revisiting interval and revisiting period, and established an optical satellite information acquisition capability model based on the target identification. This model can provide a reference for the subsequent evaluation of the effectiveness of remote sensing satellite information acquisition. Finally, we took optical remote sensing satellites, such as China Resources Satellite and GF Satellites, as examples, combined with three typical tasks of point target observation, regional target observation, and continuous observation of moving targets to verify and evaluate the proposed model.

## 2. Evaluation Model

In order to scientifically carry out the quantitative assessment of satellite resources and improve the capability of remote sensing information acquisition, this paper is to build a remote sensing observation capability evaluation model (RSOCE).The RSOCE model

includes an evaluation index system and specific calculation formulas for remote sensing satellite information acquisition capabilities to establish the quantitative relation.

The analytic hierarchy process (AHP) was developed by Thomas L. Saaty in the 1970s, and is currently used in decision making for complex scenarios. The main idea of this method is to decompose complex problems into different factors according to the nature of the problem and the overall goal to be achieved. Then referring to the subordinate relationship between these factors, the factors are combined in different levels. After constructing the logical hierarchy, decision-makers can systematically assess alternatives by comparing each selected criterion in pairs.

The AHP combines qualitative and quantitative analysis to provide a quantitative expression method. It is easy to analyze complex multi-objective problems with the analytic hierarchy process. The AHP is more suitable for decision-making problems that have hierarchical and interlaced evaluation indicators, and the target value is difficult to describe quantitatively.

Using the AHP to construct the RSOCE model are as follows.

### 2.1. Building a Hierarchical Evaluation Model

At first, we need figure out the goal and influence factors of the evaluation task, and decompose them into sub-goals and evaluation criterion factors. The RSCOE model here needs to consider the capability of observing any point target, any regional target and any moving target. These can be considered independent requirements. Therefore, the observation capability index is decomposed into three second-level indices, namely point target observation capability, regional target observation capability, and moving target observation capability. Then, each second-level evaluation index is further decomposed and refined into third-level indicators. There are interrelated influences and subordination relations among different index levels, and the evaluation values of lower-level indicators are aggregated into the evaluation values of upper-level indicators according to their weights, producing the total evaluation results of remote sensing collaborative observation capability, as shown in Figure 1. The details of indicators are described in Table 1.

The hierarchical effectiveness evaluation index system is generally constructed from two perspectives: information quality and information acquisition. Researchers have done more from the perspective of information acquisition. The literatures take system capability, economy, and reliability as the top-level indicators [18]. The system capability is further subdivided into three sub-indices: general survey, detailed survey, and emergency observation. It is measured by two indicators of observation timeliness and observation data quality. Finally, it is decomposed into specific indicators, such as regional coverage period, average revisit period, maximum revisit period, spectral coverage characteristics, and spatial resolution. This paper mainly evaluates the efficiency of satellite information acquisition from the time cost.

Point target observation capability means the information acquisition capability of the point target. This capability is further decomposed into 5 sub-indices: the total number of weekly target visits, the maximum revisiting interval, the minimum revisiting interval, the average revisiting interval, and the total number of weekly target observations. The total number of weekly target visits represents the total number of times the satellite resource set can visit the target in a week. The maximum revisit time interval represents the maximum time interval of two consecutive access point targets: The minimum revisit time interval represents the minimum time interval of two consecutive access point targets; and the average revisit time interval represents the average time interval of two consecutive visit points; The total number of weekly target observations indicates the number of opportunities to access the point target during the week.

Regional target observation capability indicates the information acquisition capability of the area target. This capability is further decomposed into 3 sub-indices: the observation frequency of the full coverage of the region, the total time of the full coverage of the region and the total time of the target observation. The observation frequency of regional total

coverage represents the observation visit frequency needed to achieve the global coverage of the opposite object. The total coverage time of the region is expressed as the time spent from the beginning time of the first visit to the planar target to the end time of the last visit to the area target when the full coverage of the region is completed. The total time of target observation is expressed as the sum of detection time of each visit to the regional target in the process of achieving full coverage of the region.

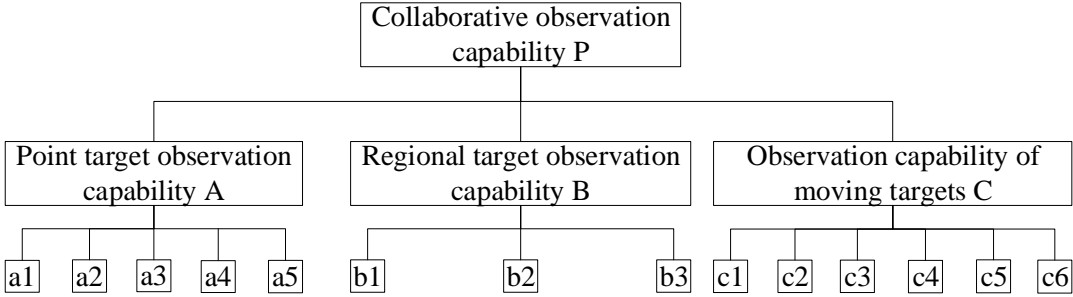

**Figure 1.** Remote sensing observation capability evaluation index system.

**Table 1.** The description of the label in the Figure 1.

| Label | Description |
|---|---|
| a1 | Total number of weekly target visits |
| a2 | Maximum revisiting interval |
| a3 | Minimum revisiting interval |
| a4 | Average revisiting interval |
| a5 | Weekly target visits |
| b1 | Observation frequency of the full coverage of the region |
| b2 | Total time of the full coverage of the region |
| b3 | Total time of the target observation |
| c1 | Total number of target observations |
| c2 | Maximum revisit interval |
| c3 | Minimum revisit interval |
| c4 | Average revisit interval |
| c5 | Total duration of target observation |
| c6 | Overall observation capability |

Observation capability of moving targets means the capability to follow and observe moving targets to obtain information. This capability is further decomposed into 6 sub-indices: the total number of target observations, the maximum revisit interval, the minimum revisit interval, the average revisit interval, the total duration of target observation, and the overall observation capability. The total number of target observations refers to the number of times that the target can be accessed during the whole movement of the target. The maximum revisit time interval represents the maximum time interval of two consecutive access point targets. The minimum revisit time interval represents the minimum time interval of two consecutive access point targets. The average revisit time interval represents the average time interval of two consecutive visit points. The total time of target observation is expressed as the sum of detection time of each visit to the moving target in the whole process of target movement. Whole-course observation capability indicates the ability to conduct whole-course observation on the target.

The remote sensing observation capability assessment model constructed by AHP is extendable, and new evaluation indices can be added at each level as needed. For example, when there are satellites with staring observation capability, the evaluation index of the "staring observation objective" can be added.

### 2.2. Determine Index Weight

The determination of the weight coefficient is the key to the analytic hierarchy process. The weight indicates the importance of each index in the set of index factors at each level. This article gives a quantitative expression based on the judgment of objective reality by experts who have been engaged in industry research for a long time. Then we constructed the priority relationship judgment matrix and the fuzzy consensus judgment matrix to determine the weight coefficient of the evaluation index. It mainly includes three main steps: constructing the priority relation judgment matrix, constructing the fuzzy consistent judgment matrix and calculating the weight set.

### 2.2.1. Construct the Priority Relation Judgment Matrix

The next step is to define the weights for each of the indices. This weighting is done through expert judgement with a group of experts, and is based on relative weighting of pairs of indices. Expert terms, such as "as important" or "much more important", are weighted in a mathematical way. The priority relationship judgment matrix reflects the priority relationship between two indicator factors related to a certain element in the previous layer at this level. It is assumed that element B in the previous level is related to elements $a_i$ and $a_j$ in the next level. When the elements $a_i$ and $a_j$ are compared with the elements B in terms of E, there are expressions, such as "... than... much more important" relative relationship. In order to determine the importance of element $a_i$ relative to $a_j$, it is necessary to establish a judgment scale expressed by the Numbers 0.1–0.9, as shown in Table 2.

**Table 2.** The importance scale of indicator factors.

| $r_{ij}$ | Define | Instructions |
|---|---|---|
| 0.5 | As important | Elements compared to the elements of $a_i$ and $a_j$, are equally important |
| 0.6 | A little important | Elements compared to the elements of $a_i$ and $a_j$,A little important |
| 0.7 | Obviously important | Elements compared to the elements of $a_i$ and $a_j$, are obviously important |
| 0.8 | Much more important | Elements compared to the elements of $a_i$ and $a_j$, are much more important |
| 0.9 | Extremely important | Elements compared to the elements of $a_i$ and $a_j$, are extremely important |
| 0.1, 0.2, 0.3, 0.4 | Inverse comparison | If the element $a_i$ is compared with element $a_j$ to get $r_{ij}$. the comparison between element $a_j$ and element $a_i$ is $r_{ij} = 1 - r_{ij}$ |

According to the contents in the table above, the relative importance of each index in the same level relative to its upper level index is judged by using the expert investigation method, including: for primary index "remote sensing observation capability", three secondary indicators, "point target observation capability", "regional target observation capability", and "moving target observation capability", between the importance of judgment. For the second-level indicator "point target observation capability", five third-level indicators, "total number of weekly target visits", "maximum revisit time interval", "minimum revisit time interval", "average revisit time interval", and "total weekly target observation time", are judged as their relative importance. For the second-level indicator "regional target observation capability", the relative importance of the three third-level indicators, "regional total coverage observation times", "regional total coverage time", and "total target observation time", is judged. For the second-level indicator "moving target observation capability", the relative importance of the six third-level indicators "total number of target observations", "maximum revisit interval", "minimum revisit interval", "average revisit interval", "total target observation duration", and "overall observation capability" was judged.

The judgment matrix composed of expert judgment values is expressed as:

$$A = \begin{bmatrix} a_{11} & a_{12} & \cdots & a_{1n} \\ a_{21} & a_{22} & \cdots & a_{2n} \\ \vdots & & & \\ a_{n1} & a_{n2} & \cdots & a_{nn} \end{bmatrix} \tag{1}$$

where, $a_{ij}$ is the importance of the index factor $A_i$ relative to $A_j$, among them, and $i$ and $j$ = 1, 2, 3,..., $n$, which represent different evaluation indices. The value on the main diagonal represents the importance of each factor in its own self-comparison, so its value is 0.5.

### 2.2.2. Construct the Fuzzy Consistent Judgment Matrix

Due to the complexity of the research problems and the differences in the understanding of the importance of the same index factors among different experts, the final judgment matrix is often inconsistent. Therefore, it is necessary to transform the fuzzy complementary matrix obtained above into a fuzzy uniform matrix. The steps of constructing the fuzzy consistency matrix are as follows:

Step 1: sum the matrix $A = (a_{ij})_{n \times n}$ by rows to get $r_i = \sum_{j=1}^n a_{ij}$, where $i$ = 1,2... $n$;

Step 2: perform the following mathematical transformation: $r_{ij} = \frac{r_i - r_j}{2(n-1)} + 0.5$

Through the above two steps can get fuzzy consistent matrix $R = (r_{ij})_{n \times n}$.

### 2.2.3. Computing Weight Set

The weight value at the level of each factor is calculated using the normalized method. The calculation steps are as follows:

Step 1: convert the complementary consistency judgment matrix A to the reciprocal judgment matrix E, $E = (e_{ij})_{n \times n}$

$$e_{ij} = \frac{a_{ij}}{a_{ji}} \ i, \ j = 1, \ 2, \ldots n. \tag{2}$$

Step 2: find the initial weight vector $A^{(0)}$

$$A^{(0)} = (a_1, a_2, \ldots, a_n)^T = \left( \frac{\sum_{j=1}^n e_{1j}}{\sum_{i=1}^n \sum_{j=1}^n e_{ij}}, \frac{\sum_{j=1}^n e_{2j}}{\sum_{i=1}^n \sum_{j=1}^n e_{ij}}, \ldots, \frac{\sum_{j=1}^n e_{nj}}{\sum_{i=1}^n \sum_{j=1}^n e_{ij}} \right)^T. \tag{3}$$

Step 3: take $A^{(0)}$ as the initial value of iteration $V_0$, use the eigenvalue method to optimize the weight vector, take $V_0 = (v_{01}, v_{02}, \ldots, v_{0n})^T$ as the initial value of iteration, use iteration $V_{k+1} = EV_k$ to find $V_{k+1}$, and find its infinite norm $\|V_{k+1}\|_\infty$.

If $\|V_{k+1}\| - \|V_k\| < \varepsilon$, then $\|V_{k+1}\|_\infty$ is the maximum eigenvalue and the iteration ends. The vector $V_{k+1}$ obtained by the normalization of I is taken as the weight vector $V_{k+1}^{(norm)}$ after optimization, namely

$$A = V_{k+1}^{norm} = \left( \frac{V_{k+1,1}}{\sum_{i=1}^n V_{k+1,i}}, \frac{V_{k+1,2}}{\sum_{i=1}^n V_{k+1,i}}, L, \frac{V_{k+1,n}}{\sum_{i=1}^n V_{k+1,i}} \right)^T. \tag{4}$$

Otherwise, take $V_k = \frac{V_{k+1}}{\|V_{k+1}\|_\infty}$ as the initial value of the new iteration and carry out iterative optimization calculation again.

Based on the above methods, the weight coefficients of each layer in the remote sensing observation capability evaluation index system are calculated. According to this method, users can adjust the weight coefficients of each layer at any time.

Firstly, through expert discussions, we have determined the values of indicators in each layer in the remote sensing observation capability evaluation index system. The details of this part are as follows.

$F_p$ is used to judge the relative importance of three secondary indices: "point target observation capability A", "regional target observation capability B", and "moving target observation capability C". The pairwise comparison among the three indicators forms a 3 × 3 matrix, $F_p$ is a specific example of Equation (1).

In this work, we were considering the observational capability of a set of satellites for disaster monitoring. We brought together a group of scholars who are directly involved in using satellite data for disaster relief, from Ministry of Emergency and China Center for Resources Satellite Data and Application. These experts individually evaluated the relative importance of the different indices, and by combining those results, as discussed above, we obtained the following matrices. The specific value of $F_p$ is as follows:

$$F_p = \begin{bmatrix} 0.5 & 0.6 & 0.7 \\ 0.4 & 0.5 & 0.6 \\ 0.3 & 0.4 & 0.5 \end{bmatrix}. \tag{5}$$

$P_A$ represents the judgment matrix of the relative importance of the 5 sub-indices of "point target observation capability A". The indices include "week visit to the total number of a1", "maximum revisit time interval a2", "minimum revisit time interval a3", "average revisit time interval a4", and "week target observation total duration a5". There is the comparison of the five indices between the two, thus forming a 5 × 5 matrix, and $P_A$ is also a specific example of Equation (1).

The specific value of $P_A$ is as follows:

$$P_A = \begin{bmatrix} 0.5 & 0.8 & 0.7 & 0.8 & 0.7 \\ 0.2 & 0.5 & 0.4 & 0.6 & 0.4 \\ 0.3 & 0.6 & 0.5 & 0.7 & 0.6 \\ 0.2 & 0.4 & 0.3 & 0.5 & 0.3 \\ 0.3 & 0.6 & 0.4 & 0.7 & 0.5 \end{bmatrix}. \tag{6}$$

$P_B$ is the judgment matrix of the relative importance of the 3 sub-indices of "regional target observation capability B". The indices include "the area covered by observation number b1", "all the area covering the time-consuming b2", and "objective observation total duration b3". The pairwise comparison among the three indicators forms a 3 × 3 matrix, and $P_B$ is a specific example of Equation (1).

The specific value of $P_B$ is as follows:

$$P_B = \begin{bmatrix} 0.5 & 0.7 & 0.7 \\ 0.3 & 0.5 & 0.5 \\ 0.3 & 0.5 & 0.5 \end{bmatrix}. \tag{7}$$

$P_C$ stands for the judgment matrix of the relative importance of the 6 sub-indices of "moving target observation capability C". The indices include " access to the total number of c1 ", "maximum revisit time interval c2", "minimum revisit time interval c3", "average revisit time interval c4", "total duration c5 target observation", and "the whole observation capability c6" judging the relative importance of each other. $P_C$ is a 6 × 6 judgment matrix.

The specific value of $P_C$ is as follows:

$$P_C = \begin{bmatrix} 0.5 & 0.7 & 0.8 & 0.8 & 0.7 & 0.9 \\ 0.3 & 0.5 & 0.7 & 0.7 & 0.6 & 0.8 \\ 0.2 & 0.3 & 0.5 & 0.6 & 0.4 & 0.7 \\ 0.2 & 0.3 & 0.4 & 0.5 & 0.4 & 0.6 \\ 0.3 & 0.4 & 0.6 & 0.6 & 0.5 & 0.7 \\ 0.1 & 0.2 & 0.3 & 0.4 & 0.3 & 0.5 \end{bmatrix}. \tag{8}$$

We sum the above matrix by rows, and transform according to $r_{ij} = \frac{r_i - r_j}{2(n-1)} + 0.5$ to form the following fuzzy consistency judgment matrix.

$$F_P^R = \begin{bmatrix} 0.5000 & 0.5500 & 0.6000 \\ 0.4500 & 0.5000 & 0.5500 \\ 0.4000 & 0.4500 & 0.5000 \end{bmatrix}, \qquad (9)$$

$$P_A^R = \begin{bmatrix} 0.5000 & 0.6400 & 0.5800 & 0.6800 & 0.6000 \\ 0.3600 & 0.5000 & 0.4400 & 0.5400 & 0.4600 \\ 0.4200 & 0.5600 & 0.5000 & 0.6000 & 0.5200 \\ 0.3200 & 0.4600 & 0.4000 & 0.5000 & 0.4200 \\ 0.4000 & 0.5400 & 0.4800 & 0.5800 & 0.5000 \end{bmatrix}, \qquad (10)$$

$$P_B^R = \begin{bmatrix} 0.5000 & 0.6000 & 0.6000 \\ 0.4000 & 0.5000 & 0.5000 \\ 0.4000 & 0.5000 & 0.5000 \end{bmatrix}, \qquad (11)$$

$$P_C^R = \begin{bmatrix} 0.5 & 0.5667 & 0.6417 & 0.6667 & 0.6083 & 0.7167 \\ 0.4443 & 0.5000 & 0.5750 & 0.6000 & 0.5417 & 0.6500 \\ 0.3583 & 0.4250 & 0.5000 & 0.5250 & 0.4667 & 0.5750 \\ 0.3333 & 0.4000 & 0.4750 & 0.5000 & 0.4417 & 0.5500 \\ 0.3917 & 0.4583 & 0.5333 & 0.5583 & 0.5000 & 0.6083 \\ 0.2833 & 0.3500 & 0.4250 & 0.4500 & 0.3917 & 0.5000 \end{bmatrix}. \qquad (12)$$

Calculate the weight according to the above weight calculation steps, and the result is shown in Figure 2.

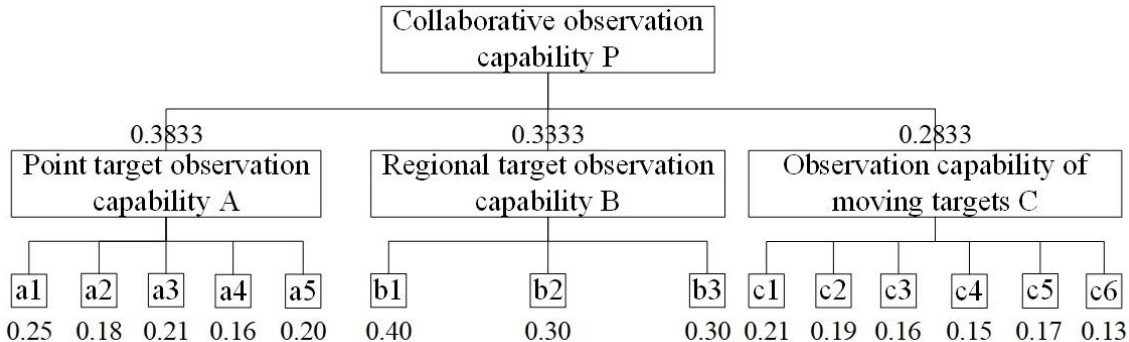

**Figure 2.** Weight allocation of evaluation indicators.

Accordingly, the evaluation formula of satellite cooperative observation capability is as follows:

Point target observation capability:

$$A = a1 \times 0.25 + a2 \times 0.18 + a3 \times 0.21 + a4 \times 0.16 + a5 \times 0.2. \qquad (13)$$

Regional target observation capability:

$$B = b1 \times 0.4 + b2 \times 0.3 + b3 \times 0.3. \qquad (14)$$

Moving target observation capability:

$$C = c1 \times 0.2133 + c2 \times 0.1867 + c3 \times 0.1567 + c4 \times 0.1467 + c5 \times 0.17 + c6 \times 0.1267. \qquad (15)$$

Satellite cooperative observation capability:

$$P = A \times 0.3833 + B \times 0.3333 + C \times 0.2833. \qquad (16)$$

### 2.3. Determine the Scoring Method of Performance Indicators

The next step is to calculate a value for the observation capability using Equation (15). For this, each satellite system needs to be given an evaluation score. The conditions for these scores were also chosen by an expert panel. Based on the above index system, the overall observation capability of satellite resources and the overall observation capability of satellite coordination are quantitatively analyzed and calculated. In order to carry out quantitative evaluation, the scoring method is determined for each value of leaf nodes performance indicators. The dimensionality normalization of various performance indicators is realized through the scoring mechanism to solve the problem of dimensional inconsistency.

The scoring method of the third-level evaluation sub-index in the above index system is given, as shown in Table 3.

**Table 3.** Quantitative scoring methods for each indicator.

| Indicators | Sub-Index | Evaluation Score |
|---|---|---|
| Point target observation capability | Total number of weekly target visits<br>Maximum revisit interval<br>Minimum revisit interval<br>Mean revisit interval<br>Total duration of weekly target observation | A score of 0.1 per visit.<br>Every second in the interval, the score is reduced by 0.0001.<br>Every second in the interval, the score is reduced by 0.0001.<br>Every second in the interval, the score is reduced by 0.0001.<br>Every second in the total duration, the score is increased by 0.1. |
| Regional target observation capability | Total coverage of the region<br><br>Full coverage of the area takes time<br>Total time of target observation | Every time the number of total coverage observations, the score is reduced by 0.01.<br>Every second in the full coverage, the score is reduced by 0.00001.<br>Every second in the total duration, the score is reduced by 0.01. |
| Moving target observation capability | Total number of target observations<br>Maximum revisit interval<br>Minimum revisit interval<br>Mean revisit interval<br>Total time of target observation<br>Whole process observation capability | A score of 0.1 per visit, the score is up to 10.<br>Every second in the interval, the score is reduced by 0.0001 and up to 10.<br>Every second in the interval, the score is reduced by 0.0001 and up to 10.<br>Every second in the interval, the score is reduced by 0.0001 and up to 10.<br>Every second in the total duration, the score is increased by 0.1 and up to 10.<br>With the ability to observe the whole process, the score is 10; if not, the score is 0. |

## 3. Experiment and Result Analysis

Resource satellites are a series of satellites dedicated to the exploration and research of earth resources in China. The resource satellite mentioned in this paper is the ZY3-2 satellite launched in 2016. Gaofen (GF) is a series of Chinese civilian remote sensing satellites for the state-sponsored program China High-definition Earth Observation System (CHEOS) [19]. The CHEOS comprises the elements of the spaceborne system, the near-space system, aerial system, the ground system and application system as a whole to realize Earth observation at high temporal, spatial, and spectral resolution. The GF satellites aim to build advanced earth observation systems for land, atmosphere and oceans, and provide services and decision-making support for modern agriculture, disaster prevention and mitigation, public security and other major areas.

Aiming at three typical observation tasks of point target observation, regional target observation and moving target accompanying, we designed three kinds of observation task determination, which compare and analyze the observation effects of using resource satellites alone and using resources satellites and GF series satellites [20]. Then, the experiment quantitatively evaluated the information acquisition capability of remote sensing satellite observation in terms of the total number of target observations per unit time, the maximum revisiting time interval, the total cumulative target observation time per unit time, the time-consuming of regional full coverage and the continuous accompanying ability of targets. Finally, based on the analysis results, the improvement of remote sensing observation capability brought by GF series and other satellites is evaluated.

### 3.1. Scenario 1: Analysis of the Improvement of Target Observation Capability

Taking satellite observation of point targets as an example, we compare and analyze the differences in the total number of weekly visits, the maximum revisiting time interval,

the minimum revisiting time interval, the average revisiting time interval and the total observation time of the weekly target under the two conditions, and illustrate the improvement of remote sensing observation capability of point targets by satellite cooperation, as shown in Table 4.

**Table 4.** Improvement of target observation capability.

| Observation Resources | | a1 | a2 | a3 | a4 | a5 |
|---|---|---|---|---|---|---|
| Resource satellite | Test | 112 | 7 h 51 min 14 s (28,274 s) | 0 | 1 h 34 min 23 s (5663 s) | 1 min 59 s (119 s) |
| | Score | 11.2 | −2.8274 | 0 | −0.5663 | 11.9 |
| Satellite synergy | Test | 167 | 6 h 29 min 3 s (23,343 s) | 0 | 58 min 35 s (4515 s) | 2 min 36 s (156 s) |
| | Score | 16.7 | −2.3343 | 0 | −0.3515 | 15.6 |

According to the point target observation capability, the quantitative evaluation formula is as follows:

Target observation capability of resource satellite points:

$$= a1 \times 0.25 + a2 \times 0.18 + a3 \times 0.21 + a4 \times 0.16 + a5 \times 0.2$$
$$= 11.2 \times 0.25 + 2.8276 \times 0.18 + 0 \times 0.21 + 0.5663 \times 0.16 + 11.9 \times 0.2$$
$$= 4.5805.$$

Satellite cooperative application point target observation capability:

$$= a1 \times 0.25 + a2 \times 0.18 + a3 \times 0.21 + a4 \times 0.16 + a5 \times 0.2$$
$$= 16.7 \times 0.25 - 2.3343 \times 0.18 + 0 \times 0.21 - 0.3515 \times 0.16 + 15.6 \times 0.2$$
$$= 6.8186.$$

The improvement of point target observation capability is as follows:

$$\frac{6.8186 - 4.5805}{4.5802} * 100\% = 49\%.$$

### 3.2. Scenario 2: Analysis of the Improvement of Regional Target Observation Capability

Regional satellite observations of a sea area is carried out, for example, by comparing the analysis using the resource satellite observations alone and at the same time use of resources and GF series satellites for collaborative observation, under the condition of two kinds of regional whole observation times, for the whole time consuming and target situation of observed total duration from three aspects, that ability of the satellite remote sensing observation on regional targets together, as shown in Table 5. Target observation is shown in Figures 3 and 4.

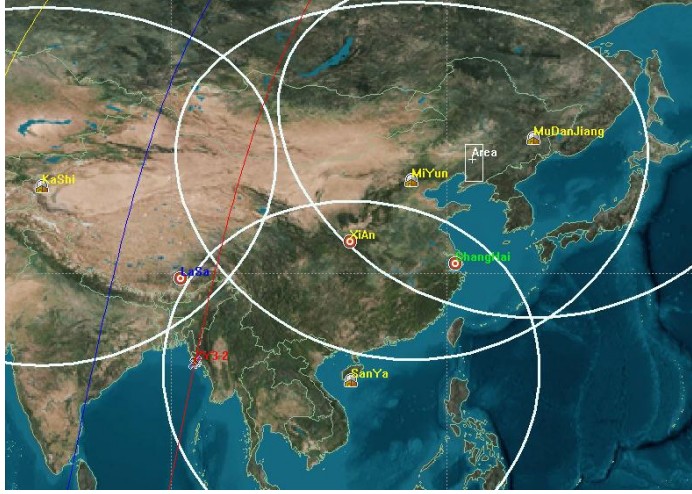

**Figure 3.** Schematic diagram of resource satellite's access to regional targets.

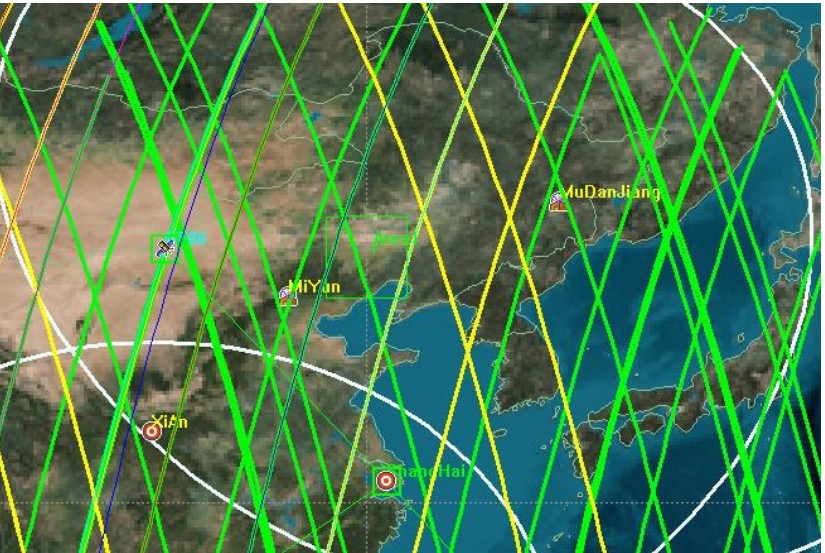

**Figure 4.** Schematic diagram of GF-resource satellite joint access to regional targets.

**Table 5.** Improvement of regional target observation capability.

| Observation Resources | | b1 | b2 | b3 |
|---|---|---|---|---|
| Resource satellite | Test | 20 | 18 h 56 min 39 s (68,199 s) | 34 min 37 s (2077 s) |
| | Score | −2 | −6.8199 | −207.7 |
| Satellite synergy | Test | 14 | 7 h 51 min 56 s (28,316 s) | 25 min 02 s (1502 s) |
| | Score | −1.4 | −2.8316 | −150.2 |

According to the point target observation capability, the quantitative evaluation formula is as follows:

To make sure the value is positive, set the initial score to 10 in both cases.

Regional target observation capability of resource satellite:

$$\begin{aligned}
&= 10 + b1 \times 0.4 + b2 \times 0.3 + b3 \times 0.3 \\
&= 10 - 0.2 \times 0.4 - 0.68199 \times 0.3 - 20.77 \times 0.3 \\
&= 3.4844.
\end{aligned}$$

Satellite coordinated regional target observation capability:

$$\begin{aligned}
&= 10 + b1 \times 0.4 + b2 \times 0.3 + b3 \times 0.3 \\
&= 10 - 0.14 \times 0.4 - 0.28316 \times 0.3 - 15.02 \times 0.3 \\
&= 5.35305.
\end{aligned}$$

The improvement of regional target observation capability is as follows:

$$\frac{5.35305 - 3.4844}{3.4844} * 100\% = 54\%.$$

Using resource satellites to observe the stripe coverage of a certain area, it takes a total of 20 visits to cover the entire waters, which takes 18 h, 56 min, and 39 s. While using resource satellites and GF series satellites to perform collaborative observation of the sea area with strip coverage, only 14 times can achieve full coverage. It takes 7 h, 51 min, and 56 s, which is 11 h, 4 min, and 43 s less than using the resource satellite observations alone.

### 3.3. Scenario 3: Analysis on the Improvement of Moving Target Observation Capability

Each satellite has its own usage characteristics, through the analysis of the characteristics of the use of satellites, it is found that the combination of some satellites with special usage modes and resource satellites can achieve an innovative way of earth observation and achieve unprecedented observation effect.

Take GF-4 satellite as an example, which is a geostationary orbit satellite with a fixed position of 105.6° E. The observation range covers the area of 7000 km × 7000 km in China's territory and surrounding areas. The single-scene imaging area covers 400 km × 400 km or more. It can be realized by satellite attitude maneuver for any position within the scope of observations, and can realize fast point to the target area. The GF-4 satellite has the ability to repeat observation for 20 s and has working modes, such as staring mode, mobile patrol mode, regional mode, and general survey mode. The payload contains a visible 50 m/medium wave infrared resolution of 400 m, more than 400 km width stare at the camera.

According to the characteristics of the GF-4 satellite mentioned above, if it can be used in conjunction with other satellites, continuous concomitant observation of large oil tankers on the surface can be formed. The specific observation steps are as follows:

(1) locate the large oil tankers, when other relevant satellites find them;
(2) according to the position of the large oil tanker determined by the observation satellite, the GF-4 satellite shall be guided to conduct image observation of the region;
(3) when the position of the large oil tanker changes, the satellite direction can be adjusted rapidly according to the new position of the large oil tanker, so as to realize the continuous companion of the large oil tanker. In addition, based on the real-time location of the large tanker, other medium and low orbit imaging satellites with access opportunities are called to observe the large tanker.

Taking the movement track of the large oil tanker passing through the western Pacific Ocean to the East China Sea as an example, this paper compares and analyzes the observation by using resources satellite alone and the observation by using resources satellite and GF series satellites at the same time. The differences in the total number of target observations, the maximum revisiting time interval, the minimum revisiting time interval, the average revisiting time interval, the total observation time of the target, and the whole observation capability under the two conditions illustrate the improvement of the remote sensing observation capability of satellites, such as high score series to moving targets, as shown in Table 6. Target observation is shown in Figures 5 and 6.

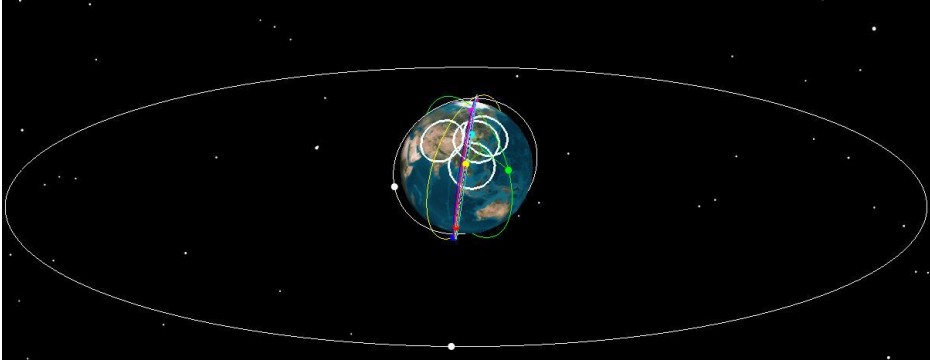

**Figure 5.** 3D diagram of the GF-resource satellite joint access to a moving target.

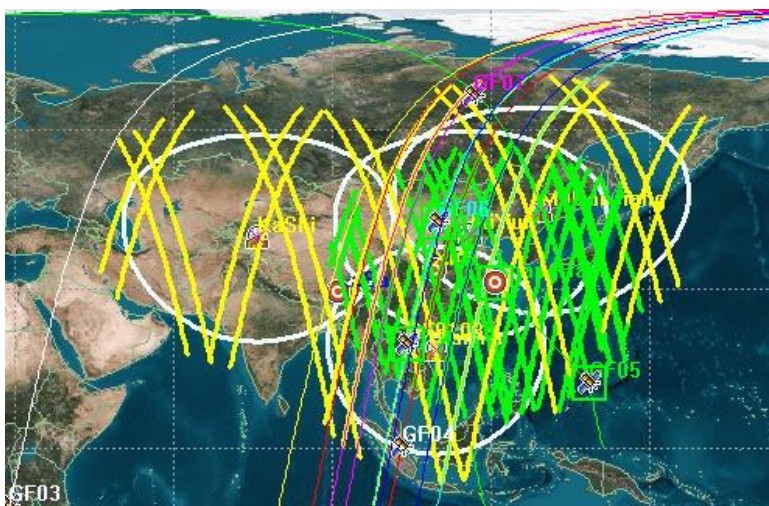

**Figure 6.** 2D diagram of the GF-resource satellite joint access to a moving target.

**Table 6.** Improvement of observation capability of moving targets.

| Observation Resources | | c1 | c2 | c3 | c4 | c5 | c6 |
|---|---|---|---|---|---|---|---|
| Resource satellite | Test | 155 | 5 h 16 min 46 s (19,006 s) | 0 | 1 h 1 min 51 s (3711 s) | 2 min 29 s (149 s) | Discontinuous |
| | Score | 15.5 | −1.9006 | 0 | −0.3711 | −0.0149 | 0 |
| GF-4 | Test | Continuous | 0 | 0 | 0 | Continuous | Continuous |
| | Score | 10 | 0 | 0 | 0 | 10 | 10 |

Resource satellite dynamic target observation capability:

$$= c1 \times 0.2133 + c2 \times 0.1867 + c3 \times 0.1567 + c4 \times 0.1467 + c5 \times 0.17 + c6 \times 0.1267$$
$$= 15.5 \times 0.2133 - 1.9006 \times 0.1867 + 0 \times 0.1567 - 0.3711 \times 0.1467 - 0.0149 \times 0.17 + 0 \times 0.1267$$
$$= 2.8943.$$

Satellite coordinated moving target observation capability:

$$= c1 \times 0.2133 + c2 \times 0.1867 + c3 \times 0.1567 + c4 \times 0.1467 + c5 \times 0.17 + c6 \times 0.1267$$
$$= 10 \times 0.2133 + 0 \times 0.1867 + 0 \times 0.1567 + 0 \times 0.1467 + 10 \times 0.17 + 10 \times 0.1267$$
$$= 5.1.$$

The improvement of dynamic target observation capability is as follows:

$$\left| \frac{5.1 - 2.8943}{2.8943} \right| * 100\% = 76\%.$$

The comparison of remote sensing observation capability between resource satellite and satellite coordination is as follows:

Resource satellite observation capability:

$$= 4.5802 \times 0.3833 + 3.4844 \times 0.3333 + 2.8943 \times 0.2833$$
$$= 3.7369.$$

Satellite cooperative observation capability:

$$= 6.8186 \times 0.3833 + 5.35305 \times 0.3333 + 5.1 \times 0.2833$$
$$= 5.8426.$$

The overall capability of remote sensing observation has been improved:

$$\frac{5.8426 - 3.7369}{3.7369} * 100\% = 56\%.$$

Using resource satellites to observe the driving process of large oil tankers, there are 155 observation opportunities during the entire movement of the large oil tanker. The length of each observation is only 1–2 s, and the cumulative observation time is only 2 min 29 s. The maximum revisit time interval is as long as 5 h, 16 min, and 46 s, and the average revisit time interval is as long as 1 h, 01 min, and 51 s. Therefore, from the point of observation effects, the moving target can be only observed intermittently, and the continuous observation effect cannot be formed.

GF-4 satellite is used to conduct accompanying observation on moving targets. The observation range of GF-4 satellite is a region of 7000 km × 7000 km in China's territory and surrounding areas, and its single-scene staring range is 400 km × 400 km. The visible light near-infrared resolution of GF-4 satellite payload is 50 m, and the mid-wave infrared resolution is 400 m. With the help of other means, satellites can find the rough orientation of large oil tankers, and guide GF-4 satellite to carry out optical image observation on large oil tankers. Due to the large oil tanker is the maximum speed of 30 knots, 56 km/hour, and the gaze range of single-scene is 400 km, it is difficult for large oil tanker to jump out of the continuous observation range of GF-4 satellite, so it can form a complete observation. The capability improvement effect brought by GF-4 satellite for moving target observation is very obvious. Observation opportunities are spread throughout the entire movement of the target, and continuous observation can be made for the whole movement of the target. The maximum revisit interval is reduced to 0, and the average revisit interval is reduced to 0.

## 4. Discussion

At present, satellite remote sensing is in an important stage with increasingly abundant means and rapid improvement in capabilities, and it is also an important period in which its benefits are becoming more prominent. The in-depth development of the information society has put forward new and higher requirements for all-round, continuous, real-time detection and rapid response capabilities. It is necessary to make a scientific assessment of the degree of satisfaction of the remote sensing satellite system.

Based on this application requirement, this paper designs a remote sensing satellite system capability index system for typical tasks, such as point target, regional target, and moving target observation. We have established an evaluation and analysis model for indicators, such as detection range, repeated detection capabilities, target identification capabilities, target positioning capabilities, and system response capabilities. On this basis, we conducted the capability assessment modeling and simulation analysis.

For the single use of resource satellite and the combined use of resource satellite and GF satellites, the observation capability of point targets increased by 49%, regional targets increased by 54%, moving targets increased by 76%, and the comprehensive observation capability increased by 56%. The test results verified the correctness of the capability index system and evaluation model.

In the future, based on the information acquisition capability index for a single imaging satellite of the same type, the information acquisition capability index can be further explored to two aspects. The first is to expand the capability elements covered by the capability index, and include capabilities such as timeliness of detection information on the basis of achieving the target acquisition volume and acquisition accuracy. The second is to expand the capability index to other types of remote sensing satellites, mainly from the detection spectrum, to infrared imaging, microwave imaging, and other remote sensing satellites, forming a serialized information acquisition capability index. Based on the expansion of these two aspects, a multi-means comprehensive information acquisition capability index is established to provide a quantitative method for comprehensive evaluation of the capability of remote sensing satellite systems.

## 5. Conclusions

Based on the new satellite capability and characteristics in recent years, this paper designs the satellite cooperative application capability evaluation model, scenario and method. We introduce the structure of the evaluation model and the functional composition of the system, the relationship between the evaluation process, and the evaluation index factors. Through the real orbit data, this paper evaluates the effect of satellite cooperative application on the improvement of remote sensing observation capability. For the single use of resource satellites and the combined use of resource satellites and GF satellites, the target's comprehensive observation capability has increased by 56%. The evaluation results show that the satellite cooperative application can significantly improve the overall remote sensing observation capability. In future work, the setting of qualitative indicators should be reduced as much as possible, and the evaluation dimension should be expanded, so as to further improve the remote sensing application model.

**Author Contributions:** Conceptualization, Z.Z.; Methodology, Z.Z., K.F. and Q.L.; Software, K.F.; Validation, Q.L.; Writing—original draft, Z.Z.; Writing—review & editing, Q.L. All authors have read and agreed to the published version of the manuscript.

**Funding:** This research received no external funding.

**Institutional Review Board Statement:** Not applicable.

**Informed Consent Statement:** Not applicable.

**Conflicts of Interest:** The authors declare no conflict of interest.

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
