# Peer review of "Evaluation Model of Remote Sensing Satellites Cooperative Observation Capability"

_remotesensing, doi:10.3390/rs13091717_

Round 1

Reviewer 1 Report

Review of the manuscript

Evaluation Model of Remote Sensing Satellites Cooperative Observation Capability

This paper proposed a new evaluation model based on analytic hierarchy process to quantitatively evaluate the capability of multi-satellite cooperative remote sensing observation.

Introduction section should be expand, you need to better prove the novelty of the article and refer to the relevant literature.

Furthermore, the format of the references in the text needs to be improved [1].

Table 1 has no numbering and title.

In Table 1, the authors presented the features taken into account in the AHP method. A comment is needed, why such features are taken into account in the analysis? What other features have been taken into account by others researchers?

The discussion section is missing, the conclusions section is only a summary of the research.

Please correct text editing.

Please include suggestions for further studies in the discussion section.

English language minor spell check required.

Reviewer 2 Report

Dear Authors, 

Before submitting your paper to a journal, you need to check its English and style.  The paper is not easy to understand and follow. So please rewrite. 

Line 70-72 In order to scientifically the satellite resources quantitative assessment on the degree of remote sensing observation ability to ascend, the project is to build a remote sensing observation ability evaluation model. 

What does this sentence mean? Did you use google translate to translate your paper? 

Line 40: comprehensive evaluation method research2. 

You should not write a citation like this. I was trying to understand what you meant by analysis1 research3 model4 and ... then found out that they are citations. 

You just have 13 citations, I don't think it is enough for journal articles. try to read more and add more to your research. 

Line 112: Regional target observation capability: refers to the observation capability of the opposite target.

This is not a style for writing a scientific article. That is more like a report. 

166: Type for index factors relative to the importance of the factors which a relative to the importance of U as the index factors, among them, i = 1, 2, 3,... , n. e = 1, 2, 3... , m.

I don't understand your English. Avoid using automatic translators. 

I am sorry I can not even find out the scientific value of your research with this English. 

Good luck

Reviewer 3 Report

Thank you for the chance to review this paper. Unfortunately, I had misunderstood the abstract when I agreed to review it and I am not an expert in "expert judgement" "analytical hierarchy" or object identification on satellites (I do look at combining satellites - but my focus is on radiometry for climate studies). 

I may, therefore, have misunderstood this paper, and I apologise for that. But I hope the attached review is helpful to you - I have worked hard to understand the paper as far as I can, and I think there are places where you could explain your method better to enable more readers to understand it. You use a lot of "jargon" and sometimes use different phrases to mean the same thing, so this makes it very hard for someone not familiar with your work to understand what you are describing.

Round 2

Reviewer 1 Report

In general, the authors try to insert the suggestions that were made and answer to all the questions in a satisfying way. I suggest to published article in present form.

Author Response

Dear reviewer,

Thanks very much for taking your time to review our manuscript. We appreciate for We are very grateful to you for affirming our work and agreeing to publish the article, and thank you again for your previous useful comments on our paper.

Best regards,

Sincerely,

Qingmei Li

Reviewer 2 Report

As I mentioned earlier, I do not understand this English. I have already asked you to re-edit it properly.  I will not understand the quality of your work when it is poorly written. 

Another example from your new version: 

line 320 : Aiming at three typical observation tasks of point target observation, regional target observation and moving target accompanying, three kinds of observation task determination are designed, which compare and analyze the observation effects of using resource satellites alone and using resources satellites and GF series satellites [20] to observe earth observation resources in terms of the total number of target visits per unit time, the maximum revisiting time interval, the total cumulative target observation time per unit time, the time-consuming of regional full coverage and the continuous accompanying ability of targets.

This paragraph is all one sentence???? What does this paragraph even mean? Just listing names. I asked you to read all and rewrite the whole paper again in proper English. This is just an example. The whole paper is written poorly. 

I don't have time and I am not qualified to edit your poor English. Sorry for this review but you need to pay attention to what you submit and what you asked already to revise. 

Good luck. 

Author Response

Dear reviewer,

Thanks very much for taking your time to review our manuscript. I really appreciate all your comments and suggestions! We are sorry that our English writing has caused you difficulty in understanding the work. We have applied for a professional language editing service.

We hope that the revised manuscript is accepted for publication in the Remote Sensing.

Best regards,

Sincerely,

Qingmei Li